# Performance Analysis of Cell-Free Massive MIMO System with Network-Assisted Full-Duplex under Time-Shifting Pilot Scheme

**Tao Ma [1], Yanfeng Hu [2,*], Zhenqi Fan [1], Xinjiang Xia [3,*] and Dongming Wang [2,*]**

[1]  State Grid Electric Power Research Institute, Nanjing 211100, China; matao2@sgepri.sgcc.com.cn (T.M.); fanzhenqi1@sgepri.sgcc.com.cn (Z.F.)
[2]  National Mobile Communications Research Laboratory, Southeast University, Nanjing 210096, China
[3]  Purple Mountain Laboratories, Nanjing 211111, China
*  Correspondence: huyanfeng@seu.edu.cn (Y.H.); xiaxinjiang@pmlabs.com.cn (X.X.); wangdm@seu.edu.cn (D.W.)

**Abstract:** A network-assisted full-duplex (NAFD) system based on a cell-free (CF) massive multiple-input, multiple-output (MIMO) framework has been proposed to satisfy the demands of higher data transmission rates and efficient communication. However, pilot contamination may occur due to the reuse of pilot sequences in a massive MIMO. With this consideration, we raise an asynchronous channel estimation method based on an uplink and downlink time-shifting pilot-sending scheme, which is able to avoid pilot sequence reuse when obtaining channel state information (CSI), while the data signals could be transmitted simultaneously at the same frequency. The transmission processes of the proposed method above are divided into three phases, including pilot phase, estimation phase, and data phase, in chronological order. When the uplink is in pilot phase, the corresponding downlink is in data phase and vice versa. After the channel state information estimation, both uplinks and downlinks are in data phase. The maximum ratio combination (MRC) receiver and the maximum ratio transmission (MRT) precoding are adopted in the uplink and downlink. The closed-form expressions are derived based on large-scale random matrix theory. We compared our asymptotic results with practical results in simulation, and find that they are well matched. Moreover, the proposed method is superior to the normal time-division duplex (TDD) system.

**Keywords:** network-assisted full-duplex; cell-free massive MIMO; time-shifting; spectral efficiency

## 1. Introduction

In communication systems, a large-scale antenna array can provide very high beamforming gains, spatial multiplexing, and spatial diversity gains, which makes massive MIMO a significant technology for the next generation of mobile communications [1,2]. Because all received pilots and data signals are processed in a central processing unit, centralized massive MIMO has a lower backhaul loss. Compared to centralized massive MIMO, the distributed massive MIMO locally processed received signals, whereas the process units are distributed randomly in a large area [3]. Based on this concept, the distribution massive MIMO system could resist the shadow fading more effectively by making full use of diversity [4]. Hence, compared with the centralized system, the distribution system could cover a larger area but has a higher backhaul loss and bad performance [5]. The cell-free massive MIMO system, as the name suggests, is a wireless communication system that is without the conception of cell and cell boundary [6]. Due to the short distance between users and receiving antennas, better communication qualities can be obtained, which makes it a research hotspot for 6G communication [7]. In comparison, the distributed access points' (APs) deployment in cell-free massive MIMO systems makes users closer to antennas, and this is conducive to a stable channel state [8–10]. Moreover, a dense antenna

deployment is adopted, which can provide higher macrodiversity gains and has a feature of channel hardening [11,12]. This structure is suitable for a user-centric system that improves the data transmission rate [6]. Basically, distributed massive MIMO and cell-free massive MIMO are two different but connected conceptions; the former has a cell boundary whereas the latter does not. However, in some special cases they are regarded as being the same. In the centralized processing scenario, if the radius of the access area is not large enough, this area becomes a cell and cell-free massive MIMO equivalent to distributed massive MIMO.

It has been proven that the system capacity can be improved by accepting the full-duplex (FD) model. Compared to the conventional half-duplex (HD) system, it may double capacity [13,14]. In [15], D. Wang has proposed an FD framework named network-assisted full-duplex (NAFD), which is mainly discussed in this paper. In this NAFD system, all the remote antenna units (RAUs) and users are randomly distributed in a large area. The Rayleigh fading model is adopted as the small-scale channel model; that is, the channel vectors are independent and obey Rayleigh distribution. In addition, as mentioned above, the centralized cell-free massive MIMO system allows both uplink and downlink baseband signals being processed at the central processing unit (CPU), which indicate that the CPU could acquire the signals in advance after precoding for all the downlink users. This reveals that it is possible to realize the DL-to-UL interference cancellation in the digital domain. By utilizing the existing HD hardware devices, an in-band FD network could be established under a cell-free massive MIMO system, which explains the concept of NAFD [15]. It cannot totally cancel the interference of transmitted RAUs and received RAUs, while basically all RAUs still work in a mode of half-duplex in our described NAFD network system. Hence, the self-interference of an FD RAU is out of our consideration. The transmission power of uplink user equipment (UEs) and transmitter RAUs should be at the same order of magnitude, which is caused by the random and even distribution of all the UEs and RAUs. Based on this assumption, the energy efficiency of an NAFD network could be further improved and the DL-to-UL interference mitigation could also be guaranteed. The spectral efficiency of an NAFD network with cell-free massive MIMO system has been analyzed in [15]. However, [15] did not consider that the great number of UEs may cause pilot contamination, which is the theme of this paper. In [16], authors derived a closed-form expression of spectral efficiency with pilot contamination. By maximizing the aggregated uplink and downlink spectral efficiency with constraints of quality-of service (QoS) and backhaul, [17] designed a method to perform joint downlink beamforming and power control for an NAFD network system. In [18], a transceiver design algorithm is studied with the consideration of maximizing the spectral efficiency while limiting the QoS and backhaul.

Pilot sequences can be used to estimate channel state information (CSI) in general, which plays an important role in the design of multi-user precoding and signal processing at the side of a base station [9]. However, as a result of limited coherent time and plenty of UEs, repeated pilot sequences are considered to be the main cause of pilot contamination [19,20]. The pilot sequences allocation schemes have been discussed in [21,22]. Nguyen et al. have proposed a novel heap-based pilot assignment algorithm that mitigates the pilot contamination while decreasing the computing time and complexity [23]. In [24], it has been proven mathematically that the performance of massive MIMO system will be affected by the pilot contamination. Both conditions with and without pilot contamination with regard to the convergences of signal-to-interference-noise (SINR) in massive MIMO system are analyzed in [25], as the number of BS antennas tend to infinity. The lower bound of a sum-rate in a massive MIMO system under pilot contamination has been derived in [26]. Considered the different UEs' channel correlation, authors in [27] present an analysis of the performance of a practical MIMO channel model, which is also available without massive scattering.

In this paper, we propose a novel method to avoid the pilot contamination. The main idea is that uplink/downlink data transmission and downlink/uplink channel estimation are simultaneous, which means the coherent interval starting time of uplink users and

downlink users are asynchronous. This is the reason that we call it a time-shifting scheme. The unique feature of this scheme is that there are always data signals transmitted in the channels during the whole coherent interval. The main contributions are summarized as follows.

- Considering the pilot contamination, a time-shifting pilot sending scheme is introduced to eliminate this problem. Compared with the scheme without a time-shifting scheme, the spectral efficiency that used our method has a significant improvement, as shown in the simulations. This provides a kind of thought that mitigates pilot contamination in a full-duplex system.
- The closed-form expressions for uplinks and downlinks approximating the sum rate of a cell-free massive MIMO NAFD system with a time-shifting pilot-sending scheme are derived based on the theory of large-scale statistics and a random matrix. The accuracy of these expressions are verified by numerical simulations in a large-scale cell-free massive MIMO system.

The organization of this paper is as follows. Section I contains the introduction and the system model is given in section II. In section III, the closed-form sum-rate expressions of this system with MRC/MRT signal processing method is derived. Numerical simulations are discussed in section IV. Finally, in section V, we conclude this paper.

The notations adopted in this paper are confirmed as follows. The vectors are denoted by lower case bold letters while the matrices are bold capitals. A bold capital letter with a superscript '-1' denotes the inverse matrix of this letter. A unit matrix with the scale of $N \times N$ is denoted by $\mathbf{I}_N$. A constant is denoted by a normal letter. The absolute value of a scalar is represented by $|\cdot|$. $\mathbb{C}^{M \times N}$ and $\mathbb{R}^{M \times N}$ denote the complex matrix (or vector) and real matrix (or vector) with a scale of $M \times N$, respectively. $[\cdot]^{\mathrm{T}}$ denotes the transpose of a matrix or vector, whereas $[\cdot]^{\mathrm{H}}$ represents the conjugate transpose one. In this paper, we use $\mathrm{tr}(\mathbf{A})$ and $\mathrm{diag}(\mathbf{a})$ to denote the trace of a square matrix $\mathbf{A}$ and a diagonal matrix with its main diagonal is formed by vector $\mathbf{a}$, respectively. $[]_{i,j}$ is the $(i, j)$th element of a matrix. Without loss of generality, $\mathrm{E}\{\cdot\}$ denotes the statistical average and $\mathrm{cov}(\cdot)$ denotes the covariance. The Hadamard product and Kronecker product are represented by $\odot$ and $\otimes$, respectively. $\mathcal{CN}(m, \sigma^2)$ denotes a circularly symmetric complex Gaussian (CSCG) distribution with a mean of $m$ and a variance of $\sigma^2$. We use $\xrightarrow{a.s.}$ to denote a convergence that almost sure (a.s.).

## 2. System Model

In this section, we propose the transmission models of both uplink and downlink. Meanwhile, the equivalent channel model with a time-shifting pilot scheme will be given in the sequel.

### 2.1. Transmission Model

The proposed NAFD network with cell-free massive MIMO architecture is depicted in Figure 1.

There are $K$ single-antenna UEs and $N$ RAUs distributed evenly in a circular area, which form this rudimentary CF massive MIMO NAFD system. We assume that the number of uplink receiver RAUs and downlink transmitting RAUs are $N_{\mathrm{u}}$ and $N_{\mathrm{d}}$, respectively. Similarly, the number of uplink UEs and downlink UEs are $K_{\mathrm{u}}$ and $K_{\mathrm{d}}$, respectively. These above parameters satisfy $N = N_{\mathrm{u}} + N_{\mathrm{d}}$ and $K = K_{\mathrm{u}} + K_{\mathrm{d}}$. Besides, we assume that all RAUs send and receive information with CPU through a backhaul network. In this backhaul network, receiver RAUs send the received signals from the uplink UEs to CPU for signal processing. After multi-user precoding, the CPU will send modulated signals to the transmitter RAUs, and then they will transmit these signals to the downlink UEs through wireless channels. In addition, the backhaul network is assumed to be perfect for simplicity, which means that this transmission channel can be error-free and of infinite capacity.

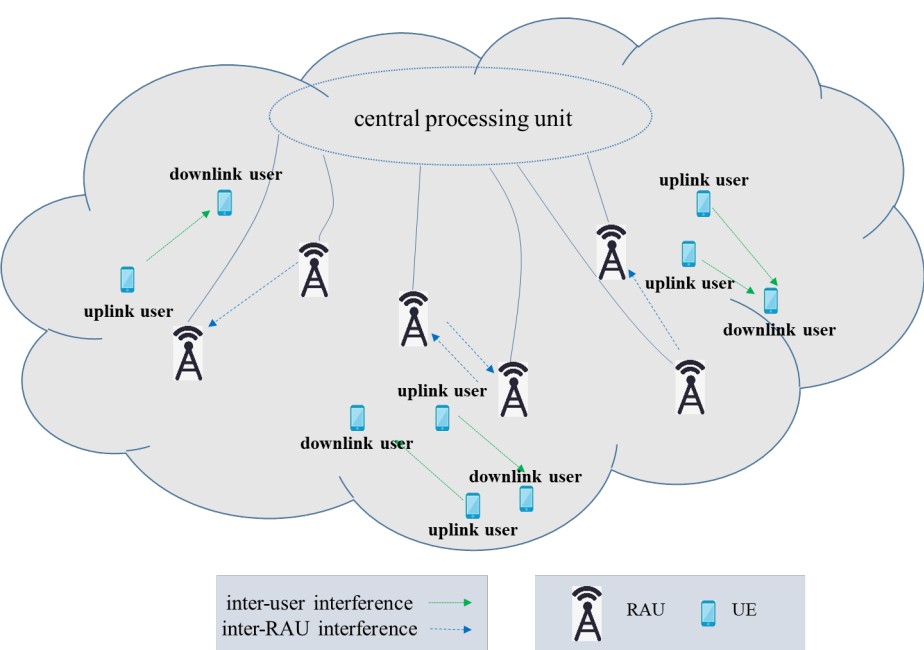

**Figure 1.** Cell-free massive MIMO NAFD system.

Our transmission scheme is illustrated in Figure 2. A complete transmission cycle is divided into pilot phase, estimation, and data phase. The pilot sequences are sent by uplink or downlink UEs in pilot phase, but not simultaneously. In the estimation phase, the CSI will be estimated according to the received pilot sequences from UEs in the base station. The data phase is regarded as the main transmission period, in which the uplink data signals will be sent to the receiver RAUs by uplink UEs, or the downlink data signals that sent by transmitter RAUs will be received by downlink UEs. Without loss of generality, we assume that the estimation phase is very short and negligible. In both the downlink and uplink channel, the same time-frequency resources are utilized to transmit signals.

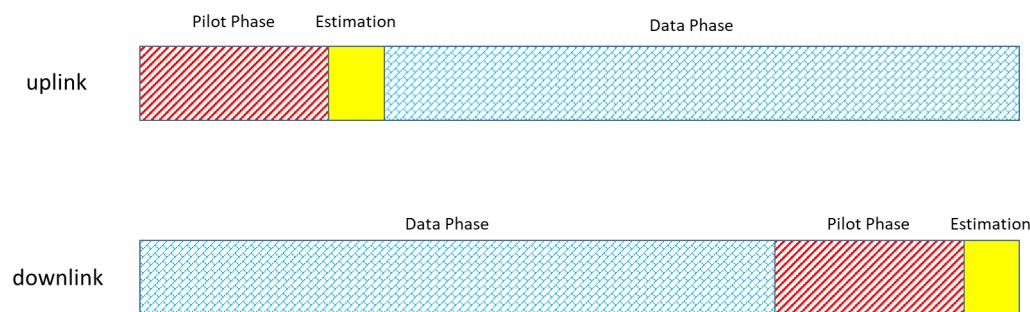

**Figure 2.** Time-shifting signal transmission scheme of NAFD system.

*2.2. Channel Model*

There are at least six kinds of channels that should be considered in our cell-free massive MIMO NAFD system. Channels between uplink UEs and receiver RAUs are denoted as $\mathbf{G}_\mathrm{u}$, whereas the channels between transmitter RAUs and downlink UEs are denoted as $\mathbf{G}_\mathrm{d}$. Two kinds of internal interference channels need to be noticed. One of them is the interuser interference channel that is denoted by $\mathbf{G}_\mathrm{iui}$ and another one is the inter-RAU interference channel that is represented by $\mathbf{G}_\mathrm{iri}$. Also, the uplink and downlink interference channels cannot be ignored. $\mathbf{G}_\mathrm{du}$ and $\mathbf{G}_\mathrm{ud}$ are used to represent the channels between downlink UEs and receive RAUs and the channels between uplink UEs and

transmitter RAUs, respectively. Assuming that each RAU has $M$ antennas, we use $\rho_{\mathrm{p}}$ to denote the transmission power of pilot sequences that are sent by UEs. Similarly, we stipulate that all uplink transmission powers are the same and denoted as $\rho_{\mathrm{u}}$, whereas the downlink transmission power is represented as $\rho_{\mathrm{d}}$.

Considering the characteristics of distributed RAUs, these channels can be expressed as:

$$
\begin{aligned}
\mathbf{G}_{\mathrm{u}} &= \mathbf{D}_{\mathrm{u}}^{1/2} \odot \mathbf{H}_{\mathrm{u}}, \\
\mathbf{G}_{\mathrm{d}} &= \mathbf{D}_{\mathrm{d}}^{1/2} \odot \mathbf{H}_{\mathrm{d}}, \\
\mathbf{G}_{\mathrm{du}} &= \mathbf{D}_{\mathrm{du}}^{1/2} \odot \mathbf{H}_{\mathrm{du}}, \\
\mathbf{G}_{\mathrm{ud}} &= \mathbf{D}_{\mathrm{ud}}^{1/2} \odot \mathbf{H}_{\mathrm{ud}}, \\
\mathbf{G}_{\mathrm{iri}} &= \mathbf{D}_{\mathrm{iri}}^{1/2} \odot \mathbf{H}_{\mathrm{iri}}, \\
\mathbf{G}_{\mathrm{iui}} &= \mathbf{D}_{\mathrm{iui}}^{1/2} \odot \mathbf{H}_{\mathrm{iui}}.
\end{aligned}
\tag{1}
$$

In these above equations, $\mathbf{D}_{\mathrm{u}}$, $\mathbf{D}_{\mathrm{d}}$, $\mathbf{D}_{\mathrm{du}}$, $\mathbf{D}_{\mathrm{ud}}$, $\mathbf{D}_{\mathrm{iri}}$, and $\mathbf{D}_{\mathrm{iui}}$ represent the large-scale fading matrices of the corresponding channels. $\mathbf{H}_{\mathrm{u}}$, $\mathbf{H}_{\mathrm{d}}$, $\mathbf{H}_{\mathrm{du}}$, $\mathbf{H}_{\mathrm{ud}}$, $\mathbf{H}_{\mathrm{iri}}$, and $\mathbf{H}_{\mathrm{iui}}$ are considered to be the circularly symmetric complex Gaussian (CSCG) matrix with zero mean unit variance and independent, identical distribution, as in small-scale fading matrices.

In channel estimation phase, the CPU estimates CSI based on the received pilot signals. We can get the received pilot sequences as following expressions:

$$
\mathbf{y}_{\mathrm{u},i} = \sqrt{\rho_{\mathrm{p}}}\,\mathbf{g}_{\mathrm{u},i} + \alpha\sqrt{\rho_{\mathrm{d}}}\,\mathbf{G}_{\mathrm{iri}}\mathbf{W}\mathbf{x}_{\mathrm{d},i} + \mathbf{z}_{\mathrm{u},i},
\tag{2}
$$

$$
\mathbf{y}_{\mathrm{d},i} = \sqrt{\rho_{\mathrm{p}}}\,\mathbf{g}_{\mathrm{d},i} + \sqrt{\rho_{\mathrm{u}}}\,\mathbf{G}_{\mathrm{ud}}\mathbf{x}_{\mathrm{u},i} + \mathbf{z}_{\mathrm{d},i},
\tag{3}
$$

where $\mathbf{g}_{\mathrm{u},i}$ is the channel vector that from the $i$-th uplink UE to receiver RAUs and $\mathbf{g}_{\mathrm{u},i} \in \mathbb{C}^{(N_{\mathrm{u}}*M)\times 1}$. $\alpha$ is the regularization factor. $\mathbf{g}_{\mathrm{d},i}$ is the channel vector that from $i$-th downlink UE to transmitter RAUs and $\mathbf{g}_{\mathrm{d},i} \in \mathbb{C}^{(N_{\mathrm{d}}*M)\times 1}$. $\mathbf{x}_{\mathrm{d},i} \in \mathbb{C}^{K_{\mathrm{d}}\times 1}$ is the downlink data signal vector and $\mathbf{W}$ is the precoding matrix. $\rho_{\mathrm{p}}$ is the pilot signal sending power, and $\mathbf{x}_{\mathrm{u},i} \in \mathbb{C}^{K_{\mathrm{u}}\times 1}$ is the uplink data signal vector. $\mathbf{z}_{\mathrm{u},i} \in \mathbb{C}^{(N_{\mathrm{u}}*M)\times 1}$ and $\mathbf{z}_{\mathrm{d},i} \in \mathbb{C}^{(N_{\mathrm{d}}*M)\times 1}$ are the noise vectors of uplink and downlink respectively. We assume that the power of received additive Gaussian complex noise of uplink and downlink are the same and with value of $\sigma^2$. According to the MMSE channel estimation method [28], the uplink channel can be estimated as

$$
\begin{aligned}
\hat{\mathbf{g}}_{u,i} &= \left[ \sqrt{\rho_p}\,\boldsymbol{\Lambda}_{u,i}\left(\rho_p\boldsymbol{\Lambda}_{u,i} + \alpha^2\rho_d\boldsymbol{\Omega}_{iri} + \sigma^2\mathbf{I}_{N_{\mathrm{u}}}\right)^{-1} \otimes \mathbf{I}_M \right] \mathbf{y}_{u,i} \\
&= \left[ \left(\sqrt{\rho_p}\,\boldsymbol{\Lambda}_{u,i}\boldsymbol{\Sigma}_{u,i}^{-1/2}\right) \otimes \mathbf{I}_M \right] \hat{\mathbf{h}}_{u,i},
\end{aligned}
\tag{4}
$$

where $\boldsymbol{\Lambda}_{\mathrm{u},i} = \mathrm{diag}(\lambda_{\mathrm{u},i,1}, \cdots, \lambda_{\mathrm{u},i,N_{\mathrm{u}}})$ and $\lambda_{\mathrm{u},i,t}$ is the large-scale fading factor that from the $i$-th uplink UE to $t$-th receiver RAU. $\boldsymbol{\Sigma}_{\mathrm{u},i} = \rho_{\mathrm{p}}\boldsymbol{\Lambda}_{\mathrm{u},i} + \alpha^2\rho_{\mathrm{d}}\boldsymbol{\Omega}_{\mathrm{iri}} + \sigma^2\mathbf{I}_{N_{\mathrm{u}}}$ where $\boldsymbol{\Omega}_{\mathrm{iri}} = \mathrm{E}\{\mathbf{G}_{\mathrm{iri}}\mathbf{W}\mathbf{W}^{\mathrm{H}}\mathbf{G}_{\mathrm{iri}}^{\mathrm{H}}\}$. We define $\hat{\mathbf{h}}_{\mathrm{u},i} = \boldsymbol{\Sigma}_{\mathrm{u},i}^{-\frac{1}{2}}\mathbf{y}_{\mathrm{u},i}$, with $\hat{\mathbf{h}}_{\mathrm{u},i} \sim \mathcal{CN}\left(0, \frac{1}{MN_{\mathrm{u}}}\mathbf{I}_{MN_{\mathrm{u}}}\right)$ and equivalent to small-scale fading.

Similarly, we can get the estimated downlink channel as $\hat{\mathbf{g}}_{\mathrm{d},i} = \left[ \left(\sqrt{\rho_p}\,\boldsymbol{\Lambda}_{\mathrm{d},i}\boldsymbol{\Sigma}_{\mathrm{d},i}^{-1/2}\right) \otimes \mathbf{I}_M \right] \hat{\mathbf{h}}_{\mathrm{d},i}$. Where $\boldsymbol{\Lambda}_{\mathrm{d},i} = \mathrm{diag}(\lambda_{\mathrm{d},i,1}, \cdots, \lambda_{\mathrm{d},i,N_{\mathrm{d}}})$ and $\lambda_{\mathrm{d},i,t}$ is the large-scale fading factor that from $i$-th downlink UE to the $t$-th transmitter RAU. $\boldsymbol{\Sigma}_{\mathrm{d},i} = \rho_{\mathrm{p}}\boldsymbol{\Lambda}_{\mathrm{d},i} + \rho_{\mathrm{u}}\boldsymbol{\Omega}_{\mathrm{ud}} + \sigma^2\mathbf{I}_{N_{\mathrm{d}}}$ where $\boldsymbol{\Omega}_{\mathrm{ud}} = \mathrm{E}\{\mathbf{G}_{\mathrm{ud}}\mathbf{G}_{\mathrm{ud}}^{\mathrm{H}}\}$. Similarly, we define $\hat{\mathbf{h}}_{\mathrm{d},l} = \boldsymbol{\Sigma}_{\mathrm{d},l}^{-\frac{1}{2}}\mathbf{y}_{\mathrm{d},l}$.

The uplink and downlink estimation error channel is defined as $\tilde{\mathbf{g}}_{\mathrm{u},i} = \mathbf{g}_{\mathrm{u},i} - \hat{\mathbf{g}}_{\mathrm{u},i}$ and $\tilde{\mathbf{g}}_{\mathrm{d},i} = \mathbf{g}_{\mathrm{d},i} - \hat{\mathbf{g}}_{\mathrm{d},i}$. We could get the channel estimation error covariance matrix as

$$
\mathrm{cov}(\tilde{\mathbf{g}}_{\mathrm{u},i}) = \left(\boldsymbol{\Lambda}_{\mathrm{u},i} - \rho_{\mathrm{p}}\boldsymbol{\Lambda}_{\mathrm{u},i}\boldsymbol{\Sigma}_{\mathrm{u},i}^{-1}\boldsymbol{\Lambda}_{\mathrm{u},i}\right) \otimes \mathbf{I}_M,
\tag{5}
$$

$$
\mathrm{cov}(\tilde{\mathbf{g}}_{\mathrm{d},i}) = \left(\boldsymbol{\Lambda}_{\mathrm{d},i} - \rho_{\mathrm{p}}\boldsymbol{\Lambda}_{\mathrm{d},i}\boldsymbol{\Sigma}_{\mathrm{d},i}^{-1}\boldsymbol{\Lambda}_{\mathrm{d},i}\right) \otimes \mathbf{I}_M.
\tag{6}
$$

### 3. Asymptotic Sum Rate

By utilizing the proposed MMSE channel estimation method, we can only obtain the imperfect CSI for the uplink and downlink channels. In this section, the MRC receiver and MRT precoding are adopted in uplink and downlink, respectively. Based on this conception, we will derive the asymptotic ergodic sum-rates of a cell-free massive MIMO NAFD system. Before that, a lemma should be introduced, which lays the foundation of the following sum-rates derivation [29].

**Lemma 1.** *Assuming that matrix* $\mathbf{A} \in \mathbb{C}^{M \times M}$ *has uniformly bounded spectral norm, vector* $\mathbf{x}, \mathbf{y} \sim \mathcal{CN}\left(0, \frac{1}{M}\mathbf{I}_M\right)$ *are independent with each other and both of them are independent with matrix* $\mathbf{A}$ *[30,31]. Then we have*

$$\mathbf{x}^H \mathbf{A} \mathbf{y} \xrightarrow[M \to \infty]{a.s.} 0, \tag{7}$$

$$\mathbf{x}^H \mathbf{A} \mathbf{x} - \frac{1}{M} Tr(\mathbf{A}) \xrightarrow[M \to \infty]{a.s.} 0. \tag{8}$$

There are two different data transmission modes: one link is in data phase and the other one is in pilot phase, or both links are in data phase. These two transmission modes are analyzed as follows.

#### 3.1. Downlink Pilot Phase and Uplink Data Phase

The received uplink signal with MRC receiver scheme can be written as

$$\mathbf{y}_{\mathrm{u}} = \hat{\mathbf{G}}_{\mathrm{u}}^H \big( \sqrt{\rho_{\mathrm{u}}} \mathbf{G}_{\mathrm{u}} \mathbf{x}_{\mathrm{u}} + \sqrt{\rho_{\mathrm{p}}} \mathbf{G}_{\mathrm{du}} \boldsymbol{\varphi}_{\mathrm{d}} + \mathbf{z}_{\mathrm{u}} \big), \tag{9}$$

where $\mathbf{x}_{\mathrm{u}}$ is the uplink data signal vector and $\boldsymbol{\varphi}_{\mathrm{d}}$ is the downlink pilot vector. $\hat{\mathbf{G}}_{\mathrm{u}} = [\hat{\mathbf{g}}_{\mathrm{u},1}, \dots, \hat{\mathbf{g}}_{\mathrm{u},K_u}]$ is denoted as the estimated uplink channel matrix. Also, $\mathbf{z}_{\mathrm{u}}$ is the noise vector with the variance of $\sigma^2$. According to this expression, the received signal of $k$-th uplink UE can be expressed as

$$
\begin{aligned}
\mathbf{y}_{\mathrm{u},k} &= \hat{\mathbf{g}}_{\mathrm{u},k}^H \big( \sqrt{\rho_{\mathrm{u}}} (\hat{\mathbf{G}}_{\mathrm{u}} + \tilde{\mathbf{G}}_{\mathrm{u}}) \mathbf{x}_{\mathrm{u}} + \sqrt{\rho_{\mathrm{p}}} \mathbf{G}_{\mathrm{du}} \boldsymbol{\varphi}_{\mathrm{d}} + \mathbf{z}_{\mathrm{u}} \big) \\
&= \sqrt{\rho_{\mathrm{u}}} \hat{\mathbf{g}}_{\mathrm{u},k}^H \hat{\mathbf{g}}_{\mathrm{u},k} \mathbf{x}_{\mathrm{u}} + \sum_{i \neq k}^{K_{\mathrm{u}}} \sqrt{\rho_{\mathrm{u}}} \hat{\mathbf{g}}_{\mathrm{u},k}^H \hat{\mathbf{g}}_{\mathrm{u},i} \mathbf{x}_{\mathrm{u}} \\
&\quad + \hat{\mathbf{g}}_{\mathrm{u},k}^H \big[ \sqrt{\rho_{\mathrm{u}}} \tilde{\mathbf{G}}_{\mathrm{u}} \mathbf{x}_{\mathrm{u}} + \sqrt{\rho_{\mathrm{p}}} \mathbf{G}_{\mathrm{du}} \boldsymbol{\varphi}_{\mathrm{d}} + \mathbf{z}_{\mathrm{u}} \big],
\end{aligned} \tag{10}
$$

where $\tilde{\mathbf{G}}_{\mathrm{u}} = \mathbf{G}_{\mathrm{u}} - \hat{\mathbf{G}}_{\mathrm{u}}$. According to (10), we could get the signal-to-interference noise-ratio (SINR) of the $k$-th uplink user as

$$\eta_{\mathrm{u},k}^{\mathrm{p}} = \frac{\rho_{\mathrm{u}} \left| \hat{\mathbf{g}}_{\mathrm{u},k}^H \hat{\mathbf{g}}_{\mathrm{u},k} \right|^2}{\sum\limits_{i \neq k}^{K_{\mathrm{u}}} \rho_{\mathrm{u}} \left| \hat{\mathbf{g}}_{\mathrm{u},k}^H \hat{\mathbf{g}}_{\mathrm{u},i} \right|^2 + \hat{\mathbf{g}}_{\mathrm{u},k}^H \boldsymbol{\Theta}_{\mathrm{u}} \hat{\mathbf{g}}_{\mathrm{u},k}}, \tag{11}$$

where the superscript p means the pilot phase. $\boldsymbol{\Theta}_{\mathrm{u}} = \mathrm{cov}\big( \sqrt{\rho_{\mathrm{u}}} \tilde{\mathbf{G}}_{\mathrm{u}} \mathbf{x}_{\mathrm{u}} + \sqrt{\rho_{\mathrm{p}}} \mathbf{G}_{\mathrm{du}}{}'_{\mathrm{d}} + \mathbf{z}_{\mathrm{u}} \big)$. According to the definition of estimated channel, we have

$$
\begin{aligned}
\hat{\mathbf{g}}_{\mathrm{u},k}^H \hat{\mathbf{g}}_{\mathrm{u},k} &= \rho_{\mathrm{p}} \hat{\mathbf{h}}_{\mathrm{u},k}^H \left\{ \left[ \left( \boldsymbol{\Lambda}_{\mathrm{u},k} \boldsymbol{\Sigma}_{\mathrm{u},k}^{-\frac{1}{2}} \right)^H \left( \boldsymbol{\Lambda}_{\mathrm{u},k} \boldsymbol{\Sigma}_{\mathrm{u},k}^{-\frac{1}{2}} \right) \right] \otimes \mathbf{I}_M \right\} \hat{\mathbf{h}}_{\mathrm{u},k} \\
&= \rho_{\mathrm{p}} \hat{\mathbf{h}}_{\mathrm{u},k}^H \left[ \left( \boldsymbol{\Lambda}_{\mathrm{u},k} \boldsymbol{\Sigma}_{\mathrm{u},k}^{-1} \boldsymbol{\Lambda}_{\mathrm{u},k} \right) \otimes \mathbf{I}_M \right] \hat{\mathbf{h}}_{\mathrm{u},k},
\end{aligned}
$$

where $\hat{\mathbf{h}}_{\mathrm{u},k} \sim \mathcal{CN}\left(0, \frac{1}{N_{\mathrm{u}}M}\mathbf{I}_{N_{\mathrm{u}}M}\right)$ and is independent with $\boldsymbol{\Lambda}_{\mathrm{u},k}\boldsymbol{\Sigma}_{\mathrm{u},k}^{-1}\boldsymbol{\Lambda}_{\mathrm{u},k}$. Using lemma 1, we can easily get

$$\hat{\mathbf{g}}_{\mathrm{u},k}^{\mathrm{H}}\hat{\mathbf{g}}_{\mathrm{u},k} \xrightarrow[N_{\mathrm{u}}M\to\infty]{a.s.} M\rho_{\mathrm{p}}\mathrm{Tr}\left\{\boldsymbol{\Lambda}_{\mathrm{u},k}\boldsymbol{\Sigma}_{\mathrm{u},k}^{-1}\boldsymbol{\Lambda}_{\mathrm{u},k}\right\}. \tag{12}$$

Similarly, we can derive that

$$\hat{\mathbf{g}}_{\mathrm{u},k}^{\mathrm{H}}\hat{\mathbf{g}}_{\mathrm{u},i}\hat{\mathbf{g}}_{\mathrm{u},i}^{\mathrm{H}}\hat{\mathbf{g}}_{\mathrm{u},k} \xrightarrow[N_{\mathrm{u}}M\to\infty]{a.s.} M\rho_{\mathrm{p}}^2\mathrm{Tr}\left\{\boldsymbol{\Lambda}_{\mathrm{u},k}\boldsymbol{\Sigma}_{\mathrm{u},k}^{-1}\boldsymbol{\Lambda}_{\mathrm{u},k}\boldsymbol{\Lambda}_{\mathrm{u},i}\boldsymbol{\Sigma}_{\mathrm{u},i}^{-1}\boldsymbol{\Lambda}_{\mathrm{u},i}\right\}, \tag{13}$$

$$\hat{\mathbf{g}}_{\mathrm{u},k}^{\mathrm{H}}\boldsymbol{\Theta}_{\mathrm{u}}\hat{\mathbf{g}}_{\mathrm{u},k} \xrightarrow[N_{\mathrm{u}}M\to\infty]{a.s.} M\rho_{\mathrm{p}}\kappa_{\mathrm{u},k}, \tag{14}$$

where

$$\begin{aligned}
\kappa_{\mathrm{u},k} &= \rho_{\mathrm{u}}\mathrm{Tr}\left\{\boldsymbol{\Lambda}_{\mathrm{u},k}\boldsymbol{\Sigma}_{\mathrm{u},k}^{-1}\boldsymbol{\Lambda}_{\mathrm{u},k}\left[\sum_{i=1}^{K_{\mathrm{u}}}\left(\boldsymbol{\Lambda}_{\mathrm{u},i} - \rho_{\mathrm{p}}\boldsymbol{\Lambda}_{\mathrm{u},i}\boldsymbol{\Sigma}_{\mathrm{u},i}^{-1}\boldsymbol{\Lambda}_{\mathrm{u},i}\right)\right]\right\} \\
&+ \rho_{\mathrm{p}}\mathrm{Tr}\left\{\boldsymbol{\Lambda}_{\mathrm{u},k}\boldsymbol{\Sigma}_{\mathrm{u},k}^{-1}\boldsymbol{\Lambda}_{\mathrm{u},k}\left[\sum_{j=1}^{K_{\mathrm{d}}}\left(\boldsymbol{\Lambda}_{\mathrm{du},j}\right)\right]\right\} \\
&+ \sigma^2\mathrm{Tr}\left\{\boldsymbol{\Lambda}_{\mathrm{u},k}\boldsymbol{\Sigma}_{\mathrm{u},k}^{-1}\boldsymbol{\Lambda}_{\mathrm{u},k}\right\},
\end{aligned} \tag{15}$$

where $\boldsymbol{\Lambda}_{\mathrm{du},j}$ is a diagonal matrix with the elements are large-scale fading of $j$-th UE. We define that

$$\xi_{\mathrm{u},k} = \mathrm{Tr}\left\{\boldsymbol{\Lambda}_{\mathrm{u},k}\boldsymbol{\Sigma}_{\mathrm{u},k}^{-1}\boldsymbol{\Lambda}_{\mathrm{u},k}\right\}, \tag{16}$$

$$\xi_{\mathrm{u},k,i} = \mathrm{Tr}\left\{\boldsymbol{\Lambda}_{\mathrm{u},k}\boldsymbol{\Sigma}_{\mathrm{u},k}^{-1}\boldsymbol{\Lambda}_{\mathrm{u},k}\boldsymbol{\Lambda}_{\mathrm{u},i}\boldsymbol{\Sigma}_{\mathrm{u},i}^{-1}\boldsymbol{\Lambda}_{\mathrm{u},i}\right\}, \tag{17}$$

and rewrite Equation (11) as follows:

$$\eta_{\mathrm{u},k}^{\mathrm{p}} \approx \frac{M\xi_{\mathrm{u},k}^2}{\sum_{k\neq i}\xi_{\mathrm{u},i,k} + \frac{\kappa_{\mathrm{u},k}}{\rho_{\mathrm{p}}\rho_{\mathrm{u}}}}. \tag{18}$$

Eventually, the uplink sum rate with downlink in pilot phase is expressed as

$$R_{\mathrm{u}}^{\mathrm{p}} = \sum_{k=1}^{K_{\mathrm{u}}}\log_2\left(1 + \eta_{\mathrm{u},k}^{\mathrm{p}}\right). \tag{19}$$

### 3.2. Uplink Pilot Phase and Downlink Data Phase

The MRT precoding scheme is adopted in downlink data transmission, whereas the received signal can be expressed as

$$\mathbf{y}_{\mathrm{d}} = \alpha\sqrt{\rho_{\mathrm{d}}}\mathbf{G}_{\mathrm{d}}^{\mathrm{H}}\mathbf{W}\mathbf{x}_{\mathrm{d}} + \sqrt{\rho_{\mathrm{p}}}\mathbf{G}_{\mathrm{iui}}\boldsymbol{\varphi}_{\mathrm{u}} + \mathbf{z}_{\mathrm{d}}. \tag{20}$$

Also, the $k$-th downlink UE received signal is

$$\mathbf{y}_{\mathrm{d},k} = \alpha\sqrt{\rho_{\mathrm{d}}}\hat{\mathbf{g}}_{\mathrm{d},k}^{\mathrm{H}}\mathbf{W}\mathbf{x}_{\mathrm{d}} + \alpha\sqrt{\rho_{\mathrm{d}}}\tilde{\mathbf{g}}_{\mathrm{d},k}^{\mathrm{H}}\mathbf{W}\mathbf{x}_{\mathrm{d}} + \sqrt{\rho_{\mathrm{p}}}\mathbf{g}_{\mathrm{iui},k}^{\mathrm{H}}\boldsymbol{\varphi}_{\mathrm{u}} + \mathbf{z}_{\mathrm{d},k}, \tag{21}$$

where $\alpha = \frac{1}{\sqrt{\mathrm{E}\{\mathrm{Tr}(\mathbf{W}\mathbf{W}^{\mathrm{H}})\}}}$ is the statistical power normalization factor and $\mathbf{W} = \hat{\mathbf{G}}_{\mathrm{d}}$. $\hat{\mathbf{G}}_{\mathrm{d}}$ is defined similarly to $\hat{\mathbf{G}}_{\mathrm{u}}$ as the estimated downlink channel matrix. Based on this equation, the SINR of $k$-th downlink UE is derived as

$$\eta_{\mathrm{d},k}^{\mathrm{p}} = \frac{\alpha^2\rho_{\mathrm{d}}\left|\hat{\mathbf{g}}_{\mathrm{d},k}^{\mathrm{H}}\hat{\mathbf{g}}_{\mathrm{d},k}\right|^2}{\alpha^2\rho_{\mathrm{d}}\phi_{\mathrm{d},k} + \rho_{\mathrm{p}}\mathbf{g}_{\mathrm{iui},k}^{\mathrm{H}}\mathbf{g}_{\mathrm{iui},k} + \sigma^2}, \tag{22}$$

where $\phi_{\mathrm{d},k} = \sum\limits_{i \neq k}^{K_{\mathrm{d}}} \left| \hat{\mathbf{g}}_{\mathrm{d},k}^{\mathrm{H}} \hat{\mathbf{g}}_{\mathrm{d},i} \right|^2 + \left| \tilde{\mathbf{g}}_{\mathrm{d},k}^{\mathrm{H}} \mathbf{W} \mathbf{W}^{\mathrm{H}} \tilde{\mathbf{g}}_{\mathrm{d},k} \right|$. Similarly, we have

$$\hat{\mathbf{g}}_{\mathrm{d},k}^{\mathrm{H}} \hat{\mathbf{g}}_{\mathrm{d},k} \xrightarrow[N_{\mathrm{d}} M \to \infty]{a.s.} M\rho_{\mathrm{p}} \mathrm{Tr}\left\{ \mathbf{\Lambda}_{\mathrm{d},k} \mathbf{\Sigma}_{\mathrm{d},k}^{-1} \mathbf{\Lambda}_{\mathrm{d},k} \right\}, \tag{23}$$

$$\hat{\mathbf{g}}_{\mathrm{d},k}^{\mathrm{H}} \hat{\mathbf{g}}_{\mathrm{d},i} \hat{\mathbf{g}}_{\mathrm{d},i}^{\mathrm{H}} \hat{\mathbf{g}}_{\mathrm{d},k} \xrightarrow[N_{\mathrm{d}} M \to \infty]{a.s.} M\rho_{\mathrm{p}}^2 \mathrm{Tr}\left\{ \mathbf{\Lambda}_{\mathrm{d},k} \mathbf{\Sigma}_{\mathrm{d},k}^{-1} \mathbf{\Lambda}_{\mathrm{d},k} \mathbf{\Lambda}_{\mathrm{d},i} \mathbf{\Sigma}_{\mathrm{d},i}^{-1} \mathbf{\Lambda}_{\mathrm{d},i} \right\}. \tag{24}$$

Mathematically, it is easy to derive that $\tilde{\mathbf{g}}_{\mathrm{d},k}^{\mathrm{H}} \hat{\mathbf{G}}_{\mathrm{d}} \hat{\mathbf{G}}_{\mathrm{d}}^{\mathrm{H}} \tilde{\mathbf{g}}_{\mathrm{d},k} = \mathrm{Tr}\left\{ \tilde{\mathbf{g}}_{\mathrm{d},k}^{\mathrm{H}} \hat{\mathbf{G}}_{\mathrm{d}} \hat{\mathbf{G}}_{\mathrm{d}}^{\mathrm{H}} \tilde{\mathbf{g}}_{\mathrm{d},k} \right\} = \mathrm{Tr}\left\{ \hat{\mathbf{G}}_{\mathrm{d}}^{\mathrm{H}} \tilde{\mathbf{g}}_{\mathrm{d},k} \tilde{\mathbf{g}}_{\mathrm{d},k}^{\mathrm{H}} \hat{\mathbf{G}}_{\mathrm{d}} \right\}$, while

$$\hat{\mathbf{G}}_{\mathrm{d}}^{\mathrm{H}} \tilde{\mathbf{g}}_{\mathrm{d},k} \tilde{\mathbf{g}}_{\mathrm{d},k}^{\mathrm{H}} \hat{\mathbf{G}}_{\mathrm{d}} \xrightarrow[N_{\mathrm{d}} M]{a.s.} \rho_{\mathrm{p}} \begin{bmatrix} \mathbf{\Omega}_{\mathrm{d},k,1} & 0 & 0 \\ 0 & \ddots & 0 \\ 0 & 0 & \mathbf{\Omega}_{\mathrm{d},k,K_{\mathrm{d}}} \end{bmatrix} \otimes \mathbf{I}_M, \tag{25}$$

and

$$\mathbf{\Omega}_{\mathrm{d},k,i} = \mathrm{Tr}\left\{ \mathbf{\Lambda}_{\mathrm{d},i} \mathbf{\Sigma}_{\mathrm{d},i}^{-1} \mathbf{\Lambda}_{\mathrm{d},i} \left( \mathbf{\Lambda}_{\mathrm{d},k} - \rho_{\mathrm{p}} \mathbf{\Lambda}_{\mathrm{d},k} \mathbf{\Sigma}_{\mathrm{d},k}^{-1} \mathbf{\Lambda}_{\mathrm{d},k} \right) \right\}. \tag{26}$$

Similar to (16) and (17), we define that

$$\xi_{\mathrm{d},k} = \mathrm{Tr}\left\{ \mathbf{\Lambda}_{\mathrm{d},k} \mathbf{\Sigma}_{\mathrm{d},k}^{-1} \mathbf{\Lambda}_{\mathrm{d},k} \right\}, \tag{27}$$

$$\xi_{\mathrm{d},k,i} = \mathrm{Tr}\left\{ \mathbf{\Lambda}_{\mathrm{d},k} \mathbf{\Sigma}_{\mathrm{d},k}^{-1} \mathbf{\Lambda}_{\mathrm{d},k} \mathbf{\Lambda}_{\mathrm{d},i} \mathbf{\Sigma}_{\mathrm{d},i}^{-1} \mathbf{\Lambda}_{\mathrm{d},i} \right\}, \tag{28}$$

$$\gamma_{\mathrm{d},k} = \sum_{i=1}^{N_{\mathrm{d}}} \lambda_{\mathrm{iui},k,i}, \tag{29}$$

where $\lambda_{\mathrm{iui},k,i}$ is the large-scale fading factor of interuser interference channel from the $k$-th UE to the $i$-th RAU. We rewrite Equation (22) as

$$\eta_{\mathrm{d},k}^{\mathrm{p}} = \frac{M\rho_{\mathrm{p}} \xi_{\mathrm{d},k}^2}{\rho_{\mathrm{p}} \sum\limits_{i \neq k}^{K_{\mathrm{d}}} \xi_{\mathrm{d},k,i} + \sum\limits_{i=1}^{K_{\mathrm{d}}} \mathbf{\Omega}_{\mathrm{d},k,i} + \frac{\rho_{\mathrm{p}} \gamma_{\mathrm{d},k} + \sigma^2}{M\alpha^2 \rho_{\mathrm{d}} \rho_{\mathrm{p}}}}. \tag{30}$$

Consequently, the closed-form expression for the downlink sum rate with uplink in the pilot phase is given as

$$R_{\mathrm{d}}^{\mathrm{p}} = \sum_{k=1}^{K_{\mathrm{d}}} \log_2\left( 1 + \eta_{\mathrm{d},k}^{\mathrm{p}} \right). \tag{31}$$

### 3.3. Both Links in Data Phase

In this mode, both the uplink and downlink are in data phase. The received signal of $k$-th uplink UE can be written as

$$\mathbf{y}_{\mathrm{u},k} = \hat{\mathbf{g}}_{\mathrm{u},k}^{\mathrm{H}} \left( \sqrt{\rho_{\mathrm{u}}} (\hat{\mathbf{G}}_{\mathrm{u}} + \tilde{\mathbf{G}}_{\mathrm{u}}) \mathbf{x}_{\mathrm{u}} + \alpha \sqrt{\rho_{\mathrm{d}}} \mathbf{G}_{\mathrm{iri}} \mathbf{W} \mathbf{x}_{\mathrm{d}} + \mathbf{z}_{\mathrm{u}} \right), \tag{32}$$

and the downlink received signal of $k$-th UE is

$$\mathbf{y}_{\mathrm{d},k} = \alpha \sqrt{\rho_{\mathrm{d}}} \hat{\mathbf{g}}_{\mathrm{d},k}^{\mathrm{H}} \mathbf{W} \mathbf{x}_{\mathrm{d}} + \alpha \sqrt{\rho_{\mathrm{d}}} \tilde{\mathbf{g}}_{\mathrm{d},k}^{\mathrm{H}} \mathbf{W} \mathbf{x}_{\mathrm{d}} + \sqrt{\rho_{\mathrm{u}}} \mathbf{g}_{\mathrm{iui},k}^{\mathrm{H}} \mathbf{x}_{\mathrm{u}} + \mathbf{z}_{\mathrm{d},k}. \tag{33}$$

We could derive the SINR of the *k*-th uplink UE and downlink UE separately:

$$\eta_{\text{u},k}^{\text{t}} = \frac{\rho_{\text{u}}\left|\hat{\mathbf{g}}_{\text{u},k}^{\text{H}}\hat{\mathbf{g}}_{\text{u},k}\right|^2}{\rho_{\text{u}}\sum_{k\neq i}\left|\hat{\mathbf{g}}_{\text{u},k}^{\text{H}}\hat{\mathbf{g}}_{\text{u},i}\right|^2 + \hat{\mathbf{g}}_{\text{u},k}^{\text{H}}\left(\hat{\mathbf{\Theta}}_{\text{u}}\right)\hat{\mathbf{g}}_{\text{u},k}}$$

$$\approx \frac{M\zeta_{\text{u},k}^2}{\sum_{k\neq i}\zeta_{\text{u},i,k} + \frac{\hat{\kappa}_{\text{u},k}}{\rho_{\text{p}}\rho_{\text{u}}}},$$

(34)

$$\eta_{\text{d},k}^{\text{t}} = \frac{\alpha^2\rho_{\text{d}}\left|\hat{\mathbf{g}}_{\text{d},k}^{\text{H}}\hat{\mathbf{g}}_{\text{d},k}\right|^2}{\alpha^2\rho_{\text{d}}\phi_{\text{d},k} + \rho_{\text{u}}\mathbf{g}_{\text{iui},k}^{\text{H}}\mathbf{g}_{\text{iui},k} + \sigma^2}$$

$$\approx \frac{M\rho_{\text{p}}\zeta_{\text{d},k}^2}{\rho_{\text{p}}\sum_{i\neq k}^{K_{\text{d}}}\zeta_{\text{d},k,i} + \sum_{i=1}^{K_{\text{d}}}\Omega_{\text{d},k,i} + \frac{\rho_{\text{u}}\hat{\gamma}_{\text{d},k}+\sigma^2}{M\alpha^2\rho_{\text{d}}\rho_{\text{p}}}},$$

(35)

where the superscript t denotes the data phase, and

$$\hat{\mathbf{\Theta}}_{\text{u}} = \text{cov}\left(\sqrt{\rho_{\text{u}}}\tilde{\mathbf{G}}_{\text{u}}\mathbf{x}_{\text{u}} + \sqrt{\rho_{\text{d}}}\alpha\mathbf{G}_{\text{iri}}\mathbf{W}\mathbf{x}_{\text{d}} + \mathbf{z}_{\text{u}}\right),$$

(36)

$$\hat{\kappa}_{\text{u},k} = \rho_{\text{u}}\cdot\text{Tr}\left\{\mathbf{\Lambda}_{\text{u},k}\mathbf{\Sigma}_{\text{u},k}^{-1}\mathbf{\Lambda}_{\text{u},k}\left[\sum_{i=1}^{K_{\text{u}}}\left(\mathbf{\Lambda}_{\text{u},i} - \rho_{\text{p}}\mathbf{\Lambda}_{\text{u},i}\mathbf{\Sigma}_{\text{u},i}^{-1}\mathbf{\Lambda}_{\text{u},i}\right)\right]\right\}$$

$$+ \alpha^2\rho_{\text{d}}\text{Tr}\left\{\mathbf{\Lambda}_{\text{u},k}\mathbf{\Sigma}_{\text{u},k}^{-1}\mathbf{\Lambda}_{\text{u},k}\left[\sum_{j=1}^{N_{\text{d}}}\tau_j\mathbf{\Lambda}_{\text{iri},j}\right]\right\}$$

$$+ \sigma^2\text{Tr}\left\{\mathbf{\Lambda}_{\text{u},k}\mathbf{\Sigma}_{\text{u},k}^{-1}\mathbf{\Lambda}_{\text{u},k}\right\},$$

(37)

$$\hat{\gamma}_{\text{d},k} = \sum_{i=1}^{K_{\text{u}}}\lambda_{\text{iui},k,i},$$

(38)

and $\tau_j$ is the *j*-th diagonal element of the matrix $\sum_{i=1}^{K_{\text{d}}}\mathbf{\Lambda}_{\text{d},i}\mathbf{\Sigma}_{\text{d},i}^{-1}\mathbf{\Lambda}_{\text{d},i}$. These equations lead to the closed-form expression of the sum rate with both links in data phase which is denoted as

$$R^t = R_u^t + R_d^t = \sum_{k=1}^{K_{\text{u}}}\log_2\left(1 + \eta_{\text{u},k}^t\right) + \sum_{i=1}^{K_d}\log_2\left(1 + \eta_{\text{d},k}^t\right).$$

(39)

If the length of pilot sequences is *L* and the length of a frame is *T*, the total sum rate can be expressed as

$$R = \frac{L}{T}\left(R_u^p + R_d^p\right) + \frac{T-2L}{T}R^t.$$

(40)

### *3.4. Summary*

At last, we summarize our scheme as the following block graph.

The first block corresponds to the uplink pilot phase and downlink data phase; the second is both the link in the data phase, and the third is the downlink pilot phase and uplink data phase. It should be noted that the order of these three blocks can be changed.

## 4. Simulations and Analysis

### *4.1. Contrasts of Theoretical and Simulated Results*

Based on the derivation in section III, the accuracy of both downlink and uplink sum rates will be validated by Monte Carlo simulations in this section. In our system, the pilot sequences are randomly assigned to each UE. We assume the coverage area of a RAU

is a circle area with radius of 1km, where the UEs and RAUs are randomly and evenly distributed. Before simulation, we establish the large-scale fading model as follows:

$$\lambda(d) = 2\bar{\lambda}\left[1 + (1 + d/d_0)^{\beta}\right]^{-1}.$$

The $\bar{\lambda}$ is stipulated as the path loss at reference point $d_0$ with $\bar{\lambda} = -34.5 - 20\log_{10}(d_0)$ dB and $d_0$=10 m. Consider that the position of RAUs should be higher than UEs, which means the shadow fading caused by terrain and bulidings will have less impact on channels between RAUs and RAUs, but more on channels between UEs and UEs. Thus, the path loss exponent is not a constant when analyzing the different channels. Consequently, $\beta_1$ denotes the pass loss exponent of the channels between UEs and RAUs with a value of 3.7, whereas $\beta_2 = 4.0$ and $\beta_3 = 3.5$ denote the channel pass loss exponent of RAUs to RAUs and UEs to UEs, respectively. In addition, in our simulated communication system, we define the system bandwidth of 10 MHz, the noise figure of 9 dB, and its power spectral density of $-174$ dB/Hz. For simplicity, the transmission power are set to be equal; that is, $\rho_p = \rho_u = \rho_d$. The simulation parameters are given in Table 1.

**Table 1.** Simulation parameters.

| Parameters | Values |
|---|---|
| Radius of a circle area $r$ | 1 km |
| Number of RAUs $N_u$ or $N_d$ | 64~512 |
| Number of UEs $K_u$ or $K_d$ | 8~64 |
| Path loss exponent of UEs to RAUs $\beta_1$ | 3.7 |
| Path loss exponent of UEs to UEs $\beta_2$ | 4.0 |
| Path loss exponent of RAUs to RAUs $\beta_3$ | 3.5 |
| Number of antennas per RAU $M$ | 8 |
| Length of pilot sequences $L$ | 70 |
| Length of frames $T$ | 700 |
| Bandwidth $B_w$ | 10 MHz |
| Noise figure | 9 dB |
| Noise power | $-174$ dBm/Hz |
| Tx power $\rho$ | 10 dBm |

As illustrated in Figure 3, the theoretical and simulated spectral efficiency are compared against the number of RAUs increasing, while the MRC/MRT signal processing scheme is adopted. In this simulation, we set the number of RAUs to vary from 64 to 448. We can see from this figure that the accuracy of asymptotic expressions (19), (31), and (39) are validated, and with the increase of the RAUs' number, the sum rates are increasing logarithmically, which means the increasing rate of spectral efficiency will be slow.

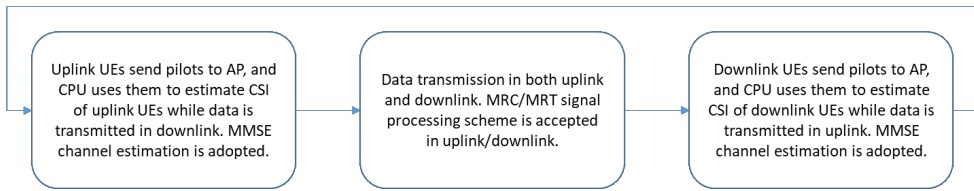

**Figure 3.** Time-shifting pilot scheme in a cell-free massive MIMO NAFD system.

Figure 4 presents a comparison of theoretical and simulated spectral efficiency against the number of UEs, where the number of UEs varies from 4 to 32. This figure reveals that the spectral efficiency increases almost linearly with the number of UEs increasing. Moreover, with the growth of number of UEs, the gap between simulation value and analysis value

becomes more and more obvious. This is because that the number of antennas is not large enough relative to the number of UEs, which leads to the lemma invalid.

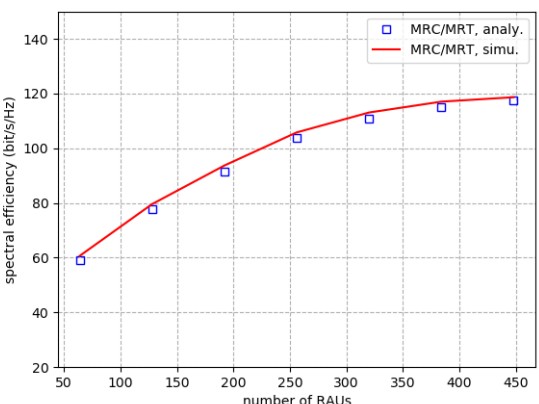

**Figure 4.** TDD-HD and NAFD spectral efficiency versus number of RAUs.

### 4.2. Contrasts of TDD System and NAFD System

In this subsection, the cell-free massive MIMO TDD system and cell-free massive MIMO NAFD system are compared with or without a time-shifting scheme in Figure 5. In this comparison, pilot contamination is adopted and pilot sequences are reused with a reuse factor equal to 2 while a time-shifting scheme is not used. We set both UEs and RAUs to be randomly and evenly distributed in an area with a radius of one kilometer.

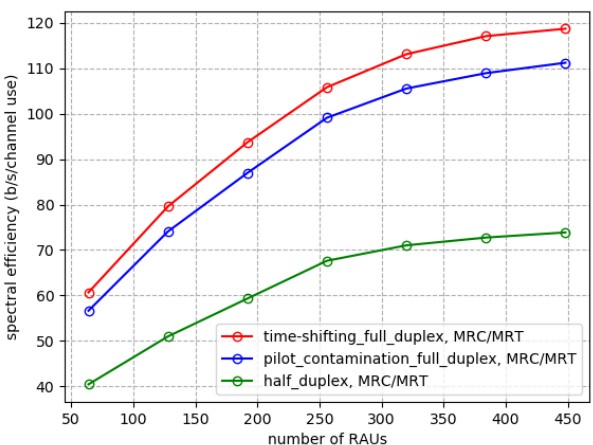

**Figure 5.** Spectral efficiency versus number of RAUs.

We could conclude from Figure 6 that by increasing the number of RAUs, the spectral efficiency of both NAFD and TDD systems will increase logarithmically. However, compared with the TDD system, the NAFD system apparently has a better performance. This indicates that a half-duplex system can be less spectral-efficient than a full-duplex. Moreover, in an NAFD network system, the time-shifting scheme has a higher spectral efficiency than pilot contamination under the same situation.

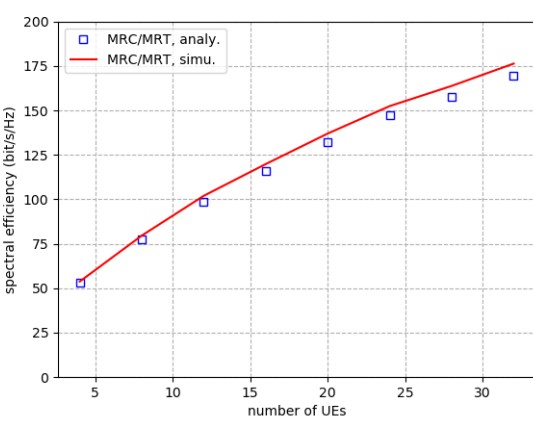

**Figure 6.** Spectral efficiency versus number of UEs.

As mentioned in the introduction, there are several methods that have been put forward to solve the pilot contamination problem. Our scheme utilizes the different processing mechanisms of uplinks and downlinks to avoid pilot contamination, which indicates that this can be only used in a full-duplex system. Moreover, the maximum reuse factor of pilot sequences is 2. If the number of UEs becomes too large, our scheme may not be helpful. NOMA and OFDM can be adopted in the wireless network system because our scheme uses the same time and frequency resources.

**5. Conclusions**

For the purpose of eliminating pilot contamination, we propose a time-shifting pilot-sending scheme that staggers pilot transmission timing in downlinks and uplinks. First, a cell-free massive MIMO NAFD network system is established, and the imperfect channel state information is estimated based on pilot sequences. The time-shifting scheme is elaborated, and based on this we derive the asymptotic expressions of downlink and uplink SINR and sum rates with the MRC/MRT signal processing method in detail. These asymptotic expressions' accuracy is verified by the numerical simulations. The spectral efficiency of a cell-free massive MIMO NAFD network system is analyzed in this paper. In the end, we give a contrast between the performance of a cell-free massive MIMO NAFD network system and a cell-free massive MIMO TDD network system, which concludes that an NAFD system should be more efficient than a TDD system and that a time-shifting scheme could improve system performance when the pilot is contaminating. However, our proposeed scheme is not elaborate enough, and there are several approaches to further improve system performance. By clustering and power allocation, the system performance can be improved by several levels, which could be the future direction of the current study.

**Author Contributions:** Conceptualization, T.M. and D.W.; writing—original draft preparation, Y.H.; writing—review and editing, T.M., Z.F., X.X. and D.W.; supervision, Y.H., Z.F., X.X. and D.W.; project administration, D.W.; All authors have read and agreed to the published version of the manuscript.

**Funding:** This work was supported by the Science and Technology Project of State Grid Corporation of China under Grant SGZJXT00JSJS2000454.

**Institutional Review Board Statement:** Not applicable.

**Informed Consent Statement:** Not applicable.

**Data Availability Statement:** Not applicable.

**Conflicts of Interest:** The authors declare no conflict of interest.

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
