# Peer review of "Performance Analysis of Cell-Free Massive MIMO System with Network-Assisted Full-Duplex under Time-Shifting Pilot Scheme"

_electronics, doi:10.3390/electronics11142171_

Round 1
Reviewer 1 Report
A network-assisted full-duplex (NAFD) system based on a cell-free (CF) massive multiple-input multiple-output (MIMO) framework has been presented to satisfy the demand for higher data transmission rate and efficient communication. The authors propose a novel method to avoid pilot contamination.
Generally, the article is well written and structured.
The abstract and introduction are appropriately written. The Introduction section gives the general framework of the research and the purpose and contribution of the manuscript.
The research has a clear objective, and the proposed theoretical model is supported in the literature.
Also, the authors have presented all intermediate steps of the processing in performance analysis, and the presentation is good.
From a technical viewpoint, the authors have satisfactorily explained the terms, and the mathematical models are well written and appropriate refereed.
At this point, an appendix with the definition-description of all the involved parameters would help the reader follow the modelling and performance description.
I would suggest the authors add a block diagram of their approach and capture in algorithmic form the steps of the processing.
The proposed approach is validated through simulations. However, the interpretation and analysis of the results aren't sufficient and should be enriched by connecting them more directly with the formulas in the theoretical analysis. Also, the authors should explain and refer to the literature about the choice of the parameter settings captured in Table 1. Please, demonstrate the environment of the experiment. Could the code of the approach be available?
A discussion section, before conclusions, is necessary to summarize the points of superiority of their approach compared to previous ones. Moreover, in the discussion section, the authors should briefly mention not only the potential issues but also the limitations of the study, emphasizing the application nature of the proposed method in practice. Due to the detailed mathematical analysis, the discussion will help the reader keep the merits and prospects of the suggested approach.
The conclusion section should be extended by discussing the future directions of the current study.
Reviewer 2 Report
I have several minor comments regarding your paper I suggest you consider revising.
1. Line 13 page 1 "simulation…" – is not clear.
2. Line 26 – suggest changing worst to bad.
3. Line 26 – "Recently…. Communication" - is not Clear.
4. Line 36 – correct typo mistake – comma.
5. Line 38 – "The small-scale…" – please revise the text.
6. Fig. 1 – text in the figure is too small or picks different fonts.
The paper is very interesting but it was hard to follow the math expressions for me.
Reviewer 3 Report
In this paper, the authors propose a new method called it time-shifting scheme to avoid the pilot contamination in a network-assisted full-duplex (NAFD) system based on cell-free (CF) massive multiple-input multiple-output (MIMO). Simulation results showed that the proposed method is superior to the normal time-division duplex (TDD) system. However, some points need to be clarified;
- Clearly explain the difference between “distribution massive MIMO” and “cell-free massive MIMO” from the perspective of system construction, pilot contamination, and channel hardening, in the introduction section.
- What is the difference between this work and reference 15? The authors have to explain hat in the introduction section.
- Discuss the possibility of using the proposed technique when massive MIMO based Full Duplex (FD) is combined with OFDM or NOMA, where mMIMO-OFDM and mMIMO-NOMA are expected to be used in the next generation of mobile communication networks as illustrated in;
"Power-Ordered NOMA with Massive MIMO for 5G Systems" Applied Sciences.
"Performance Analysis of Massive MIMO-OFDM System Incorporated with Various Transforms for Image Communication in 5G Systems" Electronics.
- Figures 3 and 4 show the simulation results and the deterministic equivalents of spectral efficiency in DL and UL, respectively. Please modify the caption of the two
figures.
- In the simulation parameters, what is the used channel model during the simulation?
- In the simulation parameters, why the pass loss exponent of the three links, i.e. UEs to RAUs, RAUs to RAUs, and UEs to UEs, are different? Explain the reason for the selection of these values.
- What is the used channel estimation technique? i.e., ideal, MMSE ...etc.
- What is the used precoding type?
Round 2
Reviewer 3 Report
Authors have done all required correction, and the article can be accepted in its current form.
This manuscript is a resubmission of an earlier submission. The following is a list of the peer review reports and author responses from that submission.